# Lung Transplantation Outcomes in Recipients Aged 70 Years or Older and the Impact of Center Volume

**DOI:** 10.3390/jcm12165372

**Published:** 2023-08-18

**Authors:** Nidhi Iyanna, Ernest G. Chan, John P. Ryan, Masashi Furukawa, Jenalee N. Coster, Chadi A. Hage, Pablo G. Sanchez

**Affiliations:** 1University of Pittsburgh School of Medicine, Pittsburgh, PA 15213, USA; iyannan2@upmc.edu; 2Division of Thoracic Surgery, Department of Cardiothoracic Surgery, University of Pittsburgh Medical Center, Pittsburgh, PA 15213, USA; chane@upmc.edu (E.G.C.); ryanjp3@upmc.edu (J.P.R.); furukawam@upmc.edu (M.F.); costerjn@upmc.edu (J.N.C.); 3Division of Pulmonary, Allergy and Critical Care Medicine, Department of Medicine, University of Pittsburgh Medical Center, Pittsburgh, PA 15213, USA; hageca@upmc.edu

**Keywords:** lung transplantation, age, elderly, center volume

## Abstract

Objective: To evaluate trends and outcomes of lung transplants (LTx) in recipients ≥ 70 years. Methods: We performed a retrospective analysis of the UNOS database identifying all patients undergoing LTx (May 2005–December 2022). Baseline characteristics and postoperative outcomes were compared by age (<70 years, ≥70 years) and center volume. Kaplan–Meier analyses were performed with pairwise comparisons between subgroups. Results: 34,957 patients underwent LTx, of which 3236 (9.3%) were ≥70 years. The rate of LTx in recipients ≥ 70 has increased over time, particularly in low-volume centers (LVCs); consequently, high-volume centers (HVCs) and LVCs perform similar rates of LTx for recipients ≥ 70. Recipients ≥ 70 had higher rates of receiving from donor after circulatory death lungs and of extended donor criteria. Recipients ≥ 70 were more likely to die of cardiovascular diseases or malignancy, while recipients < 70 of chronic primary graft failure. Survival time was shorter for recipients ≥ 70 compared to recipients < 70 old (hazard ratio (HR): 1.36, 95% confidence interval (CI): 1.28–1.44, *p* < 0.001). HVCs were associated with a survival advantage in recipients < 70 (HR: 0.91, 95% CI: 0.88–0.94, *p* < 0.001); however, in recipients ≥ 70, survival was similar between HVCs and LVCs (HR: 1.11, 95% CI: 0.99–1.25, *p* < 0.08). HVCs were more likely to perform a bilateral LTx (BLT) for obstructive lung diseases compared to LVCs, but there was no difference in BLT and single LTx likelihood for restrictive lung diseases. Conclusions: Careful consideration is needed for recipient ≥ 70 selection, donor assessment, and post-transplant care to improve outcomes. Further research should explore strategies that advance perioperative care in centers with low long-term survival for recipients ≥ 70.

## 1. Introduction

Lung transplant (LTx) is a life-saving treatment that has become the standard of care for patients with progressive end-stage lung disease. As a result of increasing life expectancy across the world, the prevalence and incidence of end-stage lung disease have dramatically grown in the elderly, resulting in a greater demand for LTx in the older adult population [1,2]. In the United States, older individuals comprise the fastest-growing segment of our population, and the proportion of candidates on the waiting list 65 years or older has nearly doubled, from 18.6% in 2011 to 33.0% in 2020 [3]. LTx in aging recipients continues to present as a complex endeavor as these individuals are particularly vulnerable to poor outcomes due to medical comorbidities and associated risk factors [4].

Historically, guidelines for the selection of LTx candidates recommended against bilateral LTx in individuals over 60 and against single LTx in those over 65 due to increased mortality in the elderly cohort [5]. More recent guidelines state that, although there is no upper age limit as an absolute contraindication for LTx, adults > 75 years old are unlikely to be suitable candidates [6]. Despite this warning, LTx in patients older than 65 years has become increasingly common [7,8] and the median recipient age has increased by over 10 years during the past three decades [9,10]. 

Recent findings from institutional studies suggest that LTx is a promising treatment option for elderly patients [7,8,11]. Improved patient survival after LTx reflects advances in surgical techniques, post-operative care, and effective immunosuppressants, indicating that advanced age alone is not necessarily an insurmountable barrier to successful LTx [12,13]. However, the increase in LTx in patients ≥ 70 years appears to have primarily occurred in high-volume centers (HVCs) [7,14]. The association between center volume and outcomes after LTx is well established with improved survival at HVCs [15,16,17]. To date, no study has examined the distribution of transplant in patients ≥ 70 years among HVCs and lower-volume centers (LVCs), nor how center volume may be associated with long-term survival in this age cohort. In the present study, we evaluated the trends and outcomes of lung transplants in recipients aged 70 years or older in the United States.

## 2. Methods

### 2.1. Study Design

National data were collected from the United Network for Organ Sharing (UNOS) standard transplant analysis and research file based on organ procurement and transplantation network data as of 31 December 2022. All patients in the study were transplanted during the lung allocation score era (May 2005–December 2022). Inclusion criteria for the study comprised all adult patients (age ≥ 18 years old) who underwent lung transplant during the study period. Exclusion criteria included multiorgan transplants. The study was approved through the University of Pittsburgh IRB protocol 20050181.

### 2.2. Data Processing

Patients who met the inclusion criteria were separated into groups based on age at time of LTx (<70 years, ≥70 years). Transplant center volume was calculated by counting the number of LTx by each center for each year of the study. Center volume was defined as HVC if 35 or more lung transplants occurred at a center during a calendar year, and LVC if fewer than 35 LTx occurred during the calendar year [18].

### 2.3. Statistical Methods

Univariable comparisons were conducted by Wilcoxon rank sum tests for continuous variables, and Pearson’s Chi-square test for categorical variables. Post hoc testing of categorical variables with more than two levels was conducted by analyzing the adjusted standardized residuals. Survival analyses were performed with a Cox regression and the Kaplan–Meier method with a log-rank test. Post-hoc Kaplan–Meier analyses were performed with pairwise comparisons between subgroups. The relationship between center volume, transplant year and proportion of transplants over the age of 70 was analyzed with generalized linear mixed modelling. Mixed modeling was conducted by regressing the age ≥ 70 variable on the transplant year, transplant volume, and the interaction between center year and center volume. Transplant year, volume and the interaction were treated as fixed effects, with center code (the identifier) as a random effect. Prior to analysis, transplant year and center volume were centered. For graphical display of the interaction between center volume and transplant year, transplant volume was dichotomized to be one standard deviation above/below the mean centered volume [19]. A Weibull regression was used to examine the association between recipient age and survival in a multivariable model. Covariates were selected based on significant differences between the age groups on donor and recipient characteristics, as well as differences in surgical procedure (e.g., transplant type).

All statistical analyses were performed with R (version 4.3.0). Univariable analyses were performed with the gtsummary package [20], and mixed modeling was performed with the GLMMAdaptive package. A *p*-value < 0.05 was considered statistically significant for all analyses.

## 3. Results

### 3.1. Sample

From the UNOS dataset, 35,986 patients met the inclusion criteria. After excluding recipients < 18 years of age (*n* = 771) and multiorgan transplants (*n* = 258), 34,957 patients were available for analysis.

### 3.2. Donor Characteristics

Recipients over the age of 70 received lungs from donors who were of older age (Table 1, 34 vs. 33 years old, *p* < 0.001), were more likely to be male (64% vs. 60%, *p* < 0.001), had slightly higher creatinine levels, and were more likely to have a pulmonary infection (68% vs. 59%, *p* < 0.001). In addition, they were more likely to have extended donor criteria, including a purulent bronchoscopy (25% vs. 23%, *p* < 0.001), an abnormal X-ray (68% vs. 59%, *p* < 0.001) and were more likely to have died of anoxia (29% vs. 24%), but less likely to have died of a cerebrovascular accident or head trauma. Age ≥ 70 recipients had higher rates of receiving donation after circulatory death (DCD) lungs than age < 70 recipients (4.8% vs. 3.7%, *p* = 0.002).

### 3.3. Recipient Characteristics

Recipients ≥ 70 years old were more likely to be male (Table 2, 75% vs. 58%, *p* < 0.001) and more likely to be White than Black or Hispanic. They were also more likely than expected to have a restrictive diagnosis (81% vs. 58%, *p* < 0.001) and less likely than expected for all other diagnosis categories.

### 3.4. Surgical and Pre-Transplant Factors

Recipients ≥ 70 years old were less likely to have pre-transplant mechanical ventilation (4.2% vs. 10%, *p* < 0.001) and had significantly lower waitlist times (33 vs. 55 days, *p* < 0.001), although they were less likely than expected to be in the ICU at time of transplant (6.8% vs. 13%, *p* < 0.001) and more likely not to be hospitalized (84% vs. 77%, *p* < 0.001) relative to recipients < 70 years of age. In addition, recipients ≥ 70 years had higher creatinine levels (0.89 mg/dL vs. 0.80 mg/dL, *p* < 0.001) but lower PA mean pressure (23 mmHg vs. 25 mmHg, *p* < 0.001) compared to recipients < 70 years. Recipients ≥ 70 were also more likely to have a history of cigarette use (68% vs. 57%, *p* < 0.001) and have had a previous cardiac surgery operation (7.4% vs. 3.6%, *p* < 0.001). Furthermore, recipients ≥ 70 years had higher than expected rates of single lung transplant (60% vs. 26%, *p* < 0.001), and were more likely to be at a HVC (79% vs. 64%, *p* < 0.001).

### 3.5. Post-Surgical Outcomes

There was a significant association between recipient age group and post-transplant ventilator duration with recipients ≥ 70 years having shorter ventilator durations (Table 3). They also had lower rates of post-transplant dialysis (5.6% vs. 7.5%), slightly shorter length of stay (16 vs. 17 days), and lower rates of treatment for rejection within one year of transplant (21% vs. 25%, *p* < 0.001). Recipients ≥ 70 had significantly lower one- and three-year survival rates (83% vs. 87% and 62% vs. 72%, respectively, *p*’s < 0.001), and there was an association between age group and cause of death (*p* < 0.001) with older recipients being more likely to die from cardiovascular and malignancy causes, but less likely to die from chronic primary graft failure (Figure 1).

### 3.6. Post-Transplant Survival Time

To investigate the association between age and survival time, patients were further stratified by center volume. There was a significant difference between groups in survival time (Figure 2, χ^2^(3) = 399, *p* < 0.001). Post hoc comparison between the four groups found that, in both HVCs and LVCs, recipients ≥ 70 had significantly shorter median survival times (HVCs: 3.7 years vs. 6.5 years; LVCs: 3.9 years vs. 5.9 years, both *p* < 0.001) than recipients < 70. In recipients < 70 years of age, HVCs had better median survival times compared to lower volume centers (6.5 years vs. 5.9 years, *p* < 0.001), but in recipients ≥ 70 years of age, there was no difference in median survival time between HVCs and LVCs (3.7 years vs. 3.9 years, *p* < 0.08). Survival analysis by Cox regression found that HVCs were associated with a survival advantage in recipients < 70 (HR: 0.91, 95% CI: 0.88–0.94, *p* < 0.001); however, in recipients ≥ 70, survival was similar between HVCs and LVCs (HR: 1.11, 95% CI: 0.99–1.25, *p* < 0.08).

### 3.7. Center Volume Effects on Recipient Characteristics and Transplant Type in Recipients ≥ 70

LVCs were more likely to have ≥70 recipients who were on average younger compared to HVC recipients (Table 4, 71 years vs. 72 years, *p* < 0.001). Although the difference in age at listing between LVCs and HVCs was statistically significant, it does not reach clinical significance due to the large sample size. LVCs had a greater proportion of ≥70 recipients of Black race compared to HVCs (4.5% vs. 2.6%, *p* = 0.026). Recipients ≥ 70 at LVCs were more likely to have pre-transplant mechanical ventilation (6.0% vs. 3.7%, *p* = 0.007), had significantly longer waitlist times (51 days vs. 29 days, *p* < 0.001), and were more likely than expected to be in the ICU at time of transplant (10% vs. 5.9%) and less likely to be not hospitalized relative to recipients at HVCs (81% vs. 85%, *p* < 0.001). LVCs were more likely to have recipients ≥ 70 with an obstructive diagnosis (21% vs. 17%) and were less likely to have recipients ≥ 70 with a restrictive diagnosis (77% vs. 82%, *p* = 0.37) compared to HVCs. There were no significant differences between recipient creatinine level, PA mean pressure, or O_2_ requirement between HVCs and LVCs, and the LAS at listing and removal were also similar. HVCs were more likely to perform a bilateral lung transplant for patients with obstructive lung disease compared to low volume centers (Figure 3, 54% vs. 44%, *p* = 0.37); however, for patients with restrictive lunge disease, the rates of bilateral lung transplants were similar for HVCs and LVCs (38% vs. 34%, *p* = 0.11).

### 3.8. Center Volume Effects over Time

In a generalized linear mixed model, transplant year, transplant volume and their interaction were all significant in predicting the probability of a recipient being ≥70 years old (Appendix A, *p* < 0.0001 for all fixed effects). In order to deconstruct the interaction term, transplant center volume was dichotomized into one standard deviation above/below the mean (corresponding to center volumes of 20 and 91 transplants per year). Examination of the interaction showed that, as transplant year increased, the proportion of transplants in recipients ≥ 70 increased in both HVCs and LVCs, but the rate increased significantly faster in LVCs to the point at which they are now performing similar rates of transplants in recipients ≥ 70 (Figure 4).

### 3.9. Multivariate Analysis

To examine the relationship between recipient age and survival, we first attempted a multivariable Cox regression. However, several variables violated the proportional hazards assumption, including the age > 70 variable. Thus, we elected to use an accelerated failure time model with Weibull regression. There was a significant association between recipient age and survival, with recipients over the age of 70 being at greater risk of death (Table 5; HR: 1.36, 95% CI: 1.28–1.44) and having 28% shorter survival time (ETR: 0.72, 95% CI: 0.67–0.76, *p* < 0.001) than recipients under the age of 70. Additional risk factors for shorter survival included lower center volume (8% shorter survival, ETR: 0.92, 95% CI: 0.88–0.95, *p* < 0.001), older donor age, higher donor creatinine, donor race other than White, having pulmonary hypertension as an indication for transplant, a history of cigarette use, prior cardiac surgery, diabetes, higher body mass index, higher creatinine, higher lung allocation score, being on mechanical ventilator at time of transplant, being hospitalized at time of transplant, use of steroids, and longer ischemic time. Recipients who were Hispanic or Other race had better survival than White recipients, and a suppurative diagnosis as indication for transplant was associated with better survival.

## 4. Discussion

The present study explores the trends, clinical outcomes, and the impact of transplant center volume on LTx in recipients aged 70 years or older. Several noteworthy findings are present in this analysis. First, the proportion of lung transplants in recipients ≥ 70 have increased during the past 10 years, with rates increasing significantly faster in LVCs; consequently, HVCs and LVCs are performing similar rates of transplants in recipients ≥ 70 in the present day. Second, there were several key differences of the ≥70 cohort, such as that recipients ≥ 70 were more likely to die of cardiovascular diseases or malignancy, while recipients < 70 were more likely to die of chronic primary graft failure. Additionally, recipients ≥ 70 had higher rates of receiving donor lungs that were categorized as extended donor criteria or from a DCD donor. Third, HVCs were associated with a survival advantage in the <70 cohort; however, this survival difference was not present in the ≥70 cohort where survival was similar between HVCs and LVCs. Of note, recipients ≥ 70 years of age had significantly shorter survival than recipients < 70 years old, independent of center volume. Advanced recipient age was also an independent risk factor for post-transplant mortality after adjusting for recipient, donor, and transplant characteristics. Finally, HVCs perform a larger proportion of bilateral lung transplants (BLT) than single lung transplants (SLT) for obstructive lung diseases compared to LVCs, but no difference in BLT and SLT likelihood was observed for restrictive lung diseases.

As the United States population ages, the proportion of LTx recipients who are ≥70 years has increased significantly over the past 10 years. This increased allocation for older patients may be attributed to the increased incidence of end-stage lung diseases, LAS change that prioritizes medical need over survival following transplant, and greater center willingness following improvements in posttransplant care [4]. Despite improvements in transplant technique and evolving postoperative care, we found that LTx recipients ≥ 70 had reduced survival time compared to their younger counterparts. After adjustment for additional risk factors of post-transplant mortality in the multivariable model, the association between recipient age and survival continued to persist, with recipients over the age of 70 being at greater risk of death. In the post-LAS era, there have been inconsistent results in long-term survival for older recipients. Some studies suggest that LTx performed in patients aged 70 years or older is associated with comparable 1-year survival to those of patients aged 60 to 69 years [7,21]. Other findings suggest that long-term outcomes of recipients ≥ 70 are inferior to younger recipients [8]. The significant mortality among recipients ≥ 70 suggest that advanced age continues to be a significant predictor of poor LTx outcomes.

Additionally, there were several key differences between donor characteristics of recipients ≥ 70 compared to recipients < 70 years. Notably, donors for age ≥ 70 recipients were more likely to be of extended donor criteria and a higher proportion were from a DCD donor. It was previously suggested that the extension of donor acceptability may increase the rate of early graft dysfunction [22]; however, recent studies have shown that extended donor criteria and DCD donations are non-inferior in terms of survival [23,24]. As the urgency and demand of LTx increases among elderly patients, utilization of extended donor criteria and DCD may alleviate the lung allograft shortage. Furthermore, there were significant differences in the cause of death for recipients ≥ 70 compared to recipients < 70 years. Older recipients were more likely to die of cardiovascular disease or malignancy while younger recipients were more likely to die to chronic primary graft failure, suggesting that post-transplant follow-up should be more tailored to prevent cardiovascular conditions and cancer rather than graft rejection in older recipients. As the debate on an upper age limit for LTx continues, careful consideration must be given to the selection of elderly recipients, assessment of appropriate donors, and tailored post-transplant care to continuously improve posttransplant outcomes.

It was previously unclear if the increased proportion of LTx recipients over the age of 70 is distributed equally amongst high and low volume centers. This study found that the rate of LTx in recipients ≥ 70 has significantly increased in LVCs over the past 10 years with HVCs and LVCs currently performing similar rates of transplants in recipients ≥ 70. This rise in caseload of recipients ≥ 70 in LVCs calls into question whether LVCs have accumulated adequate experience and are well-equipped to provide complex posttransplant care. The present study demonstrates that the long-term survival of recipients in the <70 years of age subgroup is superior in centers that perform a higher volume of lung transplants. Additionally, lower center volume was found to be an independent risk factor for shorter survival after adjustment for various recipient, donor, and transplant characteristics. This observation supports the hypothesized volume–outcome relationship in lung transplantation, where prior studies have demonstrated improved survival at HVCs compared with LVCs [15,16,17]. Yang et al. suggested one-year survival after LTx improved with increasing center volume up to as many as 33 cases per year and lower-volume centers, below 33 cases per year, had a higher risk of performing poorly [17]. However, higher center volume was not associated with a survival advantage in the ≥70 years of age cohort. Given the complex nature of LTx in older recipients, it was of interest that there was no observed volume–outcome association. This may suggest that, despite surgeons in HVCs having accumulated more transplant experience and providing advanced postoperative care, the pre-existing high risk of mortality among the elderly cohort results in similar survival outcomes independent of center volume. A previous study has suggested that HVCs have been shown to transplant older and potentially sicker patients, which may have explained the loss of survival advantage conferred by HVCs for recipients < 70 year, as LVCs may not accept high-risk elderly patients to their waitlist [17]. However, the present analysis found that recipient creatinine levels, PA mean pressure, O_2_ requirement, and the LAS at listing and removal were similar between HVCs and LVCs, suggesting that HVC recipients may not be of worse health. Of note, recipients at LVCs had significantly longer waitlist times, which may suggest that LVCs are more selective about accepting donor offers.

Our results do not imply that LVCs should not perform lung transplantations, as this would perpetuate disparities in access to care. Elimination of LVCs would force many patients to travel longer distances, and increased distance to a transplant center was found to decrease chances of being waitlisted for LTx [25]. Further research is required to explore strategies that can advance perioperative transplant care to improve survival outcomes of recipients age ≥ 70 in centers that have lower rates of long-term survival.

There is a long-standing debate over the decision to perform single or bilateral LTx in older recipients due to the added stress of a more prolonged surgery associated with BLT [26]. BLT has been positively associated with long-term survival advantage for younger patients with end-stage lung disease. However, this survival advantage is less clear in older recipients, with studies demonstrating conflicting results on long-term graft durability and mortality [27,28,29,30]. From our analysis, HVCs perform more BLT than SLT for obstructive lung diseases compared to LVCs, but similar proportions of single and BLTs for restrictive lung disease. Notably, recipients at LVCs had higher pulmonary artery mean pressures compared to recipients at HVCs. Advances in transplantation technique, post-operative care, and immunosuppression that have improved survival in elderly recipients may have led to HVCs being more likely to consider patients ≥ 70 for BLT [29].

### Limitations

We acknowledge several limitations in this study. First, as this was a retrospective analysis, there is an inherent limitation in that our results demonstrate associations rather than causality. Second, similar to other multicenter registries, the UNOS dataset is limited in the number of clinical variables that are collected, which may exclude additional factors that are relevant to explaining the differences in patients. Third, we limited our analysis to the primary outcome of long-term survival, but quality of life and functional capabilities are also important factors that should be assessed in this patient cohort.

## 5. Conclusions

The proportion of lung transplants in recipients ≥ 70 years of age has increased during the past 10 years, with rates increasing significantly faster in LVCs; consequently, HVCs and LVCs are now performing similar rates of transplants in recipients ≥ 70 years. LTx recipients ≥ 70 years had significantly shorter survival time than recipients < 70 years, suggesting that advanced age continues to be associated with poor outcomes. High volume centers were associated with a survival advantage in recipients < 70; however, survival was similar between HVCs and LVCs for LTx recipients ≥ 70 years of age. Further research is required to explore strategies that can advance perioperative transplant care to improve survival outcomes of recipients age ≥ 70 in centers that have low rates of long-term survival. Additionally, key differences of donor characteristics and cause of death of the ≥70 cohort suggest that assessment of appropriate donors and tailored post-transplant care are essential to improve posttransplant outcomes.

## Figures and Tables

**Figure 1 jcm-12-05372-f001:**
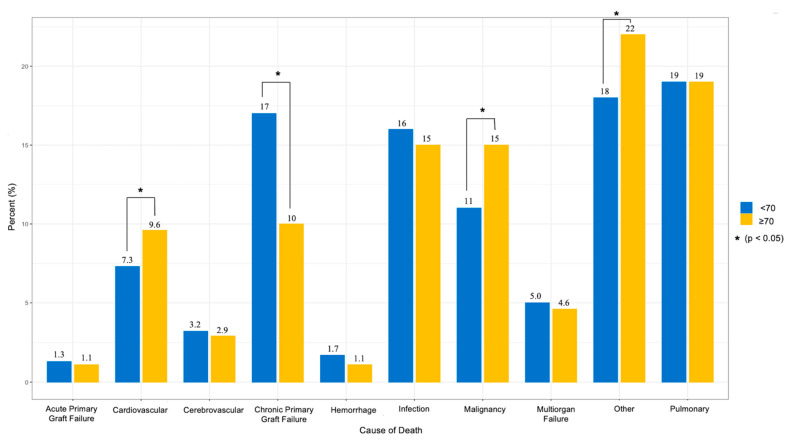
Cause of Death Comparison of Patients Above/Below Age 70.

**Figure 2 jcm-12-05372-f002:**
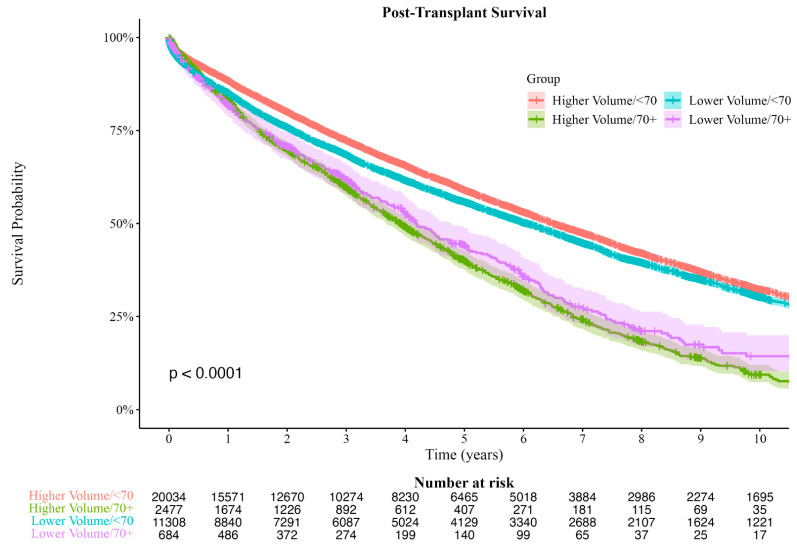
Kaplan–Meier Survival Curves Comparing Survival Probability of Patients Above/Below Age 70 by Center Volume. Shaded areas represent 95% confidence intervals.

**Figure 3 jcm-12-05372-f003:**
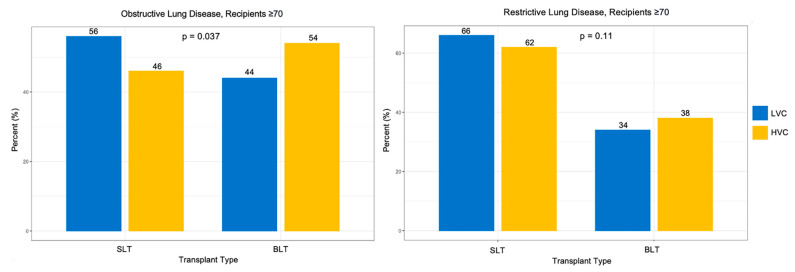
Transplant Type by Center Volume in Obstructive and Restrictive Diagnosis in Patients ≥ 70.

**Figure 4 jcm-12-05372-f004:**
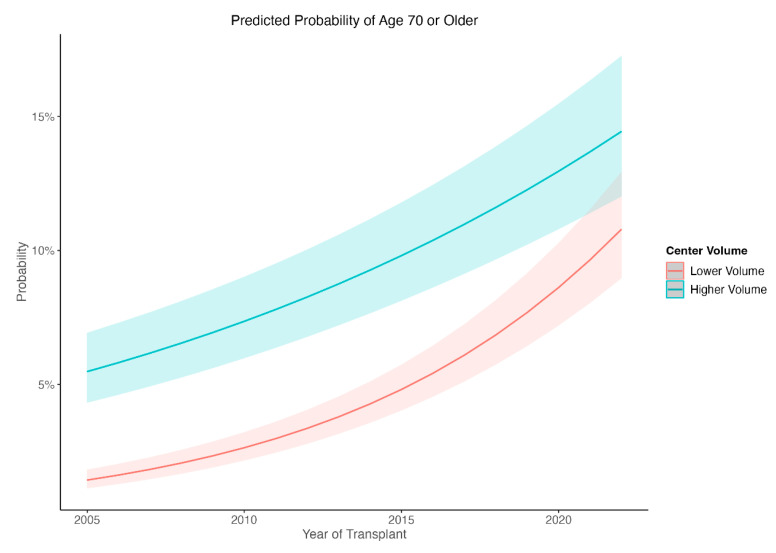
Interaction Between Center Volume and Transplant Year Predicting Likelihood of Transplant Recipient Being ≥70 Years Old.

**Table 1 jcm-12-05372-t001:** Donor Characteristics.

			Age of Recipient		
Characteristic	*N*	Overall, *N* = 34,957	<70, *N* = 31,721	≥70, *N* = 3236	*p*-Value ^†^
Age (years)	34,957	33 (23–46)	33 (23–46)	34 (24–48)	<0.001
Sex (%)	34,957				<0.001
Female		13,809 (40)	12,630 (40)	1179 (36)	
Male		21,148 (60)	19,091 (60)	2057 (64)	
Creatinine (mg/dL)	34,784	1.00 (0.73–1.40)	1.00 (0.72–1.40)	1.00 (0.77–1.55)	<0.001
Pulmonary Infection (%)	34,957	21,051 (60)	18,865 (59)	2186 (68)	<0.001
Race (%)	34,957				0.025
Black		6483 (19)	5862 (18)	621 (19)	
Hispanic		5790 (17)	5203 (16) *	587 (18) *	
Other		1344 (3.8)	1231 (3.9)	113 (3.5)	
White		21,340 (61)	19,425 (61) *	1915 (59) *	
PO_2_ (mmHg)	34,687	423 (311–490)	423 (311–490)	421 (317–488)	0.46
PaO_2_ < 300 (%)	34,686	8338 (24)	7596 (24)	742 (23)	0.18
Diabetes (%)	34,736	2643 (7.6)	2369 (7.5)	274 (8.5)	0.040
Purulent Bronchoscopy (%)	34,957	8002 (23)	7185 (23)	817 (25)	<0.001
Abnormal X-ray (%)	34,648	20,580 (59)	18,411 (59)	2169 (68)	<0.001
Cause of Death (%)	34,956				<0.001
Anoxia		8551 (24)	7615 (24) *	936 (29) *	
CNS Tumor		204 (0.6)	187 (0.6)	17 (0.5)	
CVA		10,673 (31)	9749 (31) *	924 (29) *	
Head Trauma		14,755 (42)	13,460 (42) *	1295 (40) *	
Other		773 (2.2)	709 (2.2)	64 (2.0)	
Donor Type (%)	34,956				0.002
DBD		33,643 (96)	30,561 (96)	3082 (95)	
DCD		1313 (3.8)	1159 (3.7)	154 (4.8)	

^†^ Wilcoxon rank sum test; Pearson’s Chi-squared test. * *p* < 0.05 post hoc. Data are presented as median (IQR) for continuous measures and *n* (%) for categorical measures. PO_2_, partial pressure of oxygen; PaO_2_, partial pressure of oxygen; CNS, central nervous system; CVA, cerebrovascular accidents; DCD, donation after circulatory death; DBD, donation after brain death. Pulmonary infection was confirmed by culture and reported if positive.

**Table 2 jcm-12-05372-t002:** Recipient Characteristics.

			Age of Recipient		
Characteristic	*N*	Overall, *N* = 34,957	<70, *N* = 31,721	≥70, *N* = 3236	*p*-Value ^†^
Age at Listing (years)	34,957	60 (51–65)	59 (50–64)	71 (70–73)	<0.001
Sex (%)	34,957				<0.001
Female		13,986 (40)	13,174 (42)	812 (25)	
Male		20,971 (60)	18,547 (58)	2424 (75)	
Race (%)	34,957				<0.001
Black		3145 (9.0)	3049 (9.6) *	96 (3.0) *	
Hispanic		2845 (8.1)	2670 (8.4) *	175 (5.4) *	
Other		1011 (2.9)	907 (2.9)	104 (3.2)	
White		27,956 (80)	25,095 (79) *	2861 (88) *	
Blood Type (%)	34,957				0.15
A		13,801 (39)	12,463 (39)	1338 (41)	
AB		1360 (3.9)	1236 (3.9)	124 (3.8)	
B		3891 (11)	3542 (11)	349 (11)	
O		15,905 (45)	14,480 (46)	1425 (44)	
Diagnosis Group (%)	34,957				<0.001
Obstructive		9506 (27)	8924 (28) *	582 (18) *	
Pulmonary Hypertension		1370 (3.9)	1329 (4.2) *	41 (1.3) *	
Restrictive		20,947 (60)	18,336 (58) *	2611 (81) *	
Suppurative		3134 (9.0)	3132 (9.9) *	2 (<0.1) *	
Vasodilators (%)	34,957	397 (1.1)	386 (1.2)	11 (0.3)	<0.001
Resistant Infection (%)	33,515	953 (2.8)	933 (3.1)	20 (0.6)	<0.001
Inotropes (%)	32,110	1477 (4.6)	1278 (4.4)	199 (6.4)	<0.001
History of Cigarette Use (%)	34,403	20,049 (58)	17,872 (57)	2177 (68)	<0.001
Cardiac Surgery History (%)	33,629	1341 (4.0)	1108 (3.6)	233 (7.4)	<0.001
Diabetes (%)	34,736	6859 (20)	6296 (20)	563 (18)	<0.001
Body Mass Index (kg/m^2^)	34,837	25.9 (22.1–29.2)	25.8 (21.9–29.2)	26.4 (23.8–29.0)	<0.001
PA Mean Pressure (mmHg)	32,520	25 (20–31)	25 (20–31)	23 (19–28)	<0.001
Wedge Pressure (mmHg)	32,556	10.0 (7.0–14.0)	10.0 (7.0–14.0)	9.0 (6.0–12.0)	<0.001
Cardiac Output (L/min)	32,123	5.20 (4.40–6.13)	5.20 (4.40–6.16)	5.18 (4.43–5.98)	0.040
O_2_ Requirement (L/min)	32,498	3.0 (2.0–5.0)	3.0 (2.0–5.0)	3.0 (2.0–5.0)	0.90
Creatinine (mg/dL)	34,914	0.80 (0.70–0.99)	0.80 (0.70–0.98)	0.89 (0.75–1.00)	<0.001
Waitlist Time (days)	34,957	52 (15–165)	55 (16–173)	33 (11–94)	<0.001
LAS at Listing	33,885	38 (34–46)	38 (34–46)	38 (34–45)	0.46
LAS at Removal	34,957	41 (35–53)	41 (35–54)	41 (35–50)	0.025
Pre-Tx Mechanical Ventilation (%)	34,303	3307 (9.6)	3174 (10)	133 (4.2)	<0.001
Transplant Type (%)	34,957				<0.001
Double		24,890 (71)	23,602 (74)	1288 (40)	
Single		10,067 (29)	8119 (26)	1948 (60)	
Medical Condition at Transplant (%)	34,593				<0.001
Hospitalized, Non-ICU		3331 (9.6)	3051 (9.7)	280 (8.8)	
ICU		4306 (12)	4090 (13) *	216 (6.8) *	
Not Hospitalized		26,956 (78)	24,272 (77) *	2684 (84) *	
Steroids (%)	34,070	15,527 (46)	14,305 (46)	1222 (39)	<0.001
Total Ischemic Time (hours)	33,986	5.22 (4.17–6.35)	5.25 (4.20–6.38)	4.82 (3.90–6.03)	<0.001
Center Volume (%)	34,944				<0.001
High Volume		22,870 (65)	20,329 (64)	2541 (79)	
Lower Volume		12,074 (35)	11,380 (36)	694 (21)	

^†^ Wilcoxon rank sum test; Pearson’s Chi-squared test. * *p* < 0.05 post hoc. Data are presented as median (IQR) for continuous measures and *n* (%) for categorical measures. PA, pulmonary artery, O_2_, oxygen; LAS, lung allocation score; Pre-Tx, pre-transplantation; ICU, intensive care unit.

**Table 3 jcm-12-05372-t003:** Transplant Outcomes.

			Age of Recipient		
Outcome	*N*	Overall, *N* = 34,957	<70, *N* = 31,721	≥70, *N* = 3236	*p*-Value ^†^
Post-Tx Ventilator Support (%)	33,743				<0.001
None		1058 (3.1)	950 (3.1)	108 (3.4)	
Ventilator support for ≤48 h		19,945 (59)	17,852 (58) *	2093 (67) *	
Ventilator support for >48 h but <5 days		5634 (17)	5174 (17) *	460 (15) *	
Ventilator support ≥ 5 days		7106 (21)	6630 (22) *	476 (15) *	
Airway Dehiscence (%)	34,164	525 (1.5)	480 (1.5)	45 (1.4)	0.58
Stroke (%)	34,249	811 (2.4)	733 (2.4)	78 (2.5)	0.69
PGD3 at 72 h (%)	34,957	3488 (10.0)	3140 (9.9)	348 (11)	0.12
Dialysis (Post-Tx) (%)	34,418	2531 (7.4)	2354 (7.5)	177 (5.6)	<0.001
Length of Stay (days)	34,021	17 (12–29)	17 (12–29)	16 (11–27)	<0.001
Treated for Rejection in Year 1 (%)	27,243	6704 (25)	6223 (25)	481 (21)	<0.001
Recipient Cause of Death (%)	16,044				<0.001
Acute Primary Graft Failure		205 (1.3)	189 (1.3)	16 (1.1)	
Cardiovascular		1212 (7.6)	1067 (7.3) *	145 (9.6) *	
Cerebrovascular		505 (3.1)	461 (3.2)	44 (2.9)	
Chronic Primary Graft Failure		2635 (16)	2480 (17) *	155 (10) *	
Hemorrhage		260 (1.6)	243 (1.7)	17 (1.1)	
Infection		2617 (16)	2387 (16)	230 (15)	
Malignancy		1825 (11)	1601 (11) *	224 (15) *	
Multiorgan Failure		794 (4.9)	725 (5.0)	69 (4.6)	
Other		2909 (18)	2579 (18) *	330 (22) *	
Pulmonary		3082 (19)	2796 (19)	286 (19)	
One-Year Survival (%)	32,316	28,146 (87)	25,789 (87)	2357 (83)	<0.001
Three-Year Survival (%)	27,375	19,502 (71)	18,164 (72)	1338 (62)	<0.001

^†^ Pearson’s Chi-squared test; Wilcoxon rank sum test. * *p* < 0.05 post hoc. Data are presented as median (IQR) for continuous measures and *n* (%) for categorical measures. Post-Tx, post-transplantation. PGD3, primary graft dysfunction. One-year survival and three-year survival are raw calculations of survival, not estimates.

**Table 4 jcm-12-05372-t004:** Recipient Characteristics by Center Volume in Recipients ≥ 70.

Characteristic	Overall, *N* = 3235	Lower Volume, *N* = 694	High Volume, *N* = 2541	*p*-Value ^†^
Age at Listing (years) ^‡^	72 (2)	71 (2)	72 (2)	<0.001
Race (%)				0.026
Black	96 (3.0)	31 (4.5) *	65 (2.6) *	
Hispanic	175 (5.4)	40 (5.8)	135 (5.3)	
Other	104 (3.2)	16 (2.3)	88 (3.5)	
White	2860 (88)	607 (87)	2253 (89)	
Blood Type (%)				0.31
A	1338 (41)	303 (44)	1035 (41)	
AB	124 (3.8)	20 (2.9)	104 (4.1)	
B	349 (11)	71 (10)	278 (11)	
O	1424 (44)	300 (43)	1124 (44)	
Diagnosis Group (%)				0.037
Obstructive	581 (18)	144 (21) *	437 (17) *	
Pulmonary Hypertension	41 (1.3)	12 (1.7)	29 (1.1)	
Restrictive	2611 (81)	537 (77) *	2074 (82) *	
Suppurative	2 (<0.1)	1 (0.1)	1 (<0.1)	
Sex (%)				0.14
Female	811 (25)	189 (27)	622 (24)	
Male	2424 (75)	505 (73)	1919 (76)	
Vasodilators (%)	11 (0.3)	0 (0)	11 (0.4)	0.14
Resistant Infection (%)	20 (0.6)	5 (0.7)	15 (0.6)	0.78
Inotropes (%)	199 (6.4)	17 (2.6)	182 (7.5)	<0.001
History of Cigarette Use (%)	2176 (68)	461 (67)	1715 (68)	0.49
Cardiac Surgery History (%)	233 (7.4)	42 (6.2)	191 (7.7)	0.18
Diabetes (%)	563 (18)	140 (20)	423 (17)	0.034
Body Mass Index (kg/m^2^)	26.4 (23.8–29.0)	26.2 (23.7–29.2)	26.4 (23.8–29.0)	0.77
PA Mean Pressure (mmHg)	23 (19–28)	24 (19–28)	23 (19–28)	0.18
Wedge Pressure (mmHg)	9.0 (6.0–12.0)	10.0 (6.0–13.0)	9.0 (6.0–12.0)	<0.001
Cardiac Output (L/min)	5.19 (4.43–5.98)	5.17 (4.46–6.00)	5.19 (4.43–5.96)	0.56
O_2_ Requirement (L/min)	3.0 (2.0–5.0)	3.0 (2.0–5.0)	3.0 (2.0–5.0)	0.35
Creatinine (mg/dL)	0.89 (0.75–1.00)	0.88 (0.72–1.00)	0.89 (0.75–1.00)	0.23
Days on Waitlist (days)	32 (11–94)	51 (17–131)	29 (9–83)	<0.001
LAS at Listing	38 (34–45)	38 (34–45)	38 (34–45)	0.58
LAS at Removal	41 (35–50)	41 (35–51)	41 (36–50)	0.60
Pre-Tx Mechanical Ventilation (%)	133 (4.2)	41 (6.0)	92 (3.7)	0.007
Transplant Type (%)				0.043
Double	1287 (40)	253 (36)	1034 (41)	
Single	1948 (60)	441 (64)	1507 (59)	
Medical Condition at Transplant (%)				<0.001
Hospitalized, Non-ICU	280 (8.8)	62 (9.1)	218 (8.7)	
ICU	216 (6.8)	70 (10) *	146 (5.9) *	
Not Hospitalized	2683 (84)	552 (81)	2131 (85)	
Steroids (%)	1222 (39)	318 (47)	904 (36)	<0.001
Total Ischemic Time (hours)	4.82 (3.90–6.03)	4.60 (3.62–5.70)	4.89 (4.00–6.13)	<0.001

^†^ Wilcoxon rank sum test; Pearson’s Chi-squared test; Fisher’s exact test. * *p* < 0.05 post hoc. Data are presented as median (IQR) for continuous measures unless indicated with ^‡^ where mean (SD) was used instead. For categorical measures, data are presented as *n* (%). PA, pulmonary artery, O_2_, oxygen; LAS, lung allocation score; Pre-Tx, pre-transplantation; ICU, intensive care unit.

**Table 5 jcm-12-05372-t005:** Multivariable Analysis Predicting Post-Transplant Mortality.

	b	SE	HR (95% CI)	ETR (95% CI)	*p*
Recipient Age					
<70 years	-	-	-	-	-
70+ years	−0.33	0.03	1.36 (1.28–1.44)	0.72 (0.67–0.76)	<0.001
Center Volume					
High Volume	-	-	-	-	-
Lower Volume	−0.09	0.02	1.08 (1.05–1.12)	0.92 (0.88–0.95)	<0.001
Donor Characteristics					
Age	0.00	0.00	1.00 (1.00–1.01)	1.00 (0.99–1.00)	<0.001
Sex					
Female	-	-	-	-	-
Male	0.03	0.02	0.97 (0.94–1.01)	1.03 (0.99–1.07)	0.19
Creatinine	−0.02	0.01	1.01 (1.00–1.03)	0.98 (0.97–1.00)	0.009
Pulmonary Infection					
No	-	-	-	-	-
Yes	0.02	0.02	0.99 (0.95–1.02)	1.02 (0.98–1.06)	0.42
Race					
White	-	-	-	-	-
Black	−0.17	0.02	1.17 (1.12–1.22)	0.84 (0.80–0.88)	<0.001
Hispanic	−0.08	0.03	1.08 (1.03–1.13)	0.91 (0.87–0.97)	0.002
Other	−0.14	0.05	1.13 (1.04–1.24)	0.87 (0.79–0.96)	0.01
PO_2_	0.00	0.00	1.00 (1.00–1.00)	1.00 (1.00–1.00)	0.62
Diabetes					
No	-	-	-	-	-
Yes	−0.05	0.04	1.04 (0.98–1.12)	0.95 (0.89–1.03)	0.20
Purulent Bronchoscopy					
No	-	-	-	-	-
Yes	−0.01	0.02	1.01 (0.97–1.05)	0.99 (0.94–1.03)	0.54
Abnormal Chest X-Ray					
No	-	-	-	-	-
Yes	0.03	0.02	0.97 (0.94–1.01)	1.03 (0.99–1.07)	0.15
Donation Type					
Brain Death (DBD)	-	-	-	-	-
Circulatory Death (DCD)	−0.08	0.06	1.07 (0.96–1.19)	0.93 (0.83–1.04)	0.20
Recipient Characteristics					
Race					
White	-	-	-	-	-
Black	0.01	0.03	0.99 (0.93–1.05)	1.01 (0.94–1.08)	0.820
Hispanic	0.12	0.04	0.89 (0.83–0.96)	1.13 (1.04–1.22)	0.002
Other	0.13	0.06	0.89 (0.79–0.99)	1.14 (1.01–1.28)	0.03
Diagnosis					
Obstructive	-	-	-	-	-
Pulmonary Hypertension	−0.12	0.06	1.12 (1.01–1.25)	0.88 (0.79–0.99)	0.032
Restrictive	0.05	0.03	0.95 (0.91–1.00)	1.05 (1.00–1.11)	0.06
Suppurative	0.34	0.05	0.73 (0.67–0.80)	1.41 (1.27–1.55)	<0.001
Sex					
Female	-	-	-	-	-
Male	−0.04	0.02	1.04 (1.00–1.08)	0.96 (0.92–1.00)	0.07
Vasodilators					
No	-	-	-	-	-
Yes	0.12	0.11	0.90 (0.74–1.09)	1.12 (0.91–1.38)	0.27
Resistant Infection					
No	-	-	-	-	-
Yes	−0.12	0.07	1.11 (0.98–1.26)	0.89 (0.78–1.02)	0.09
Inotropes/Vasodilators					
No	-	-	-	-	-
Yes	−0.05	0.05	1.05 (0.96–1.14)	0.95 (0.87–1.04)	0.28
History of Cigarette Use					
No	-	-	-	-	-
Yes	−0.09	0.02	1.09 (1.04–1.13)	0.91 (0.87–0.95)	<0.001
Prior Cardiac Surgery					
No	-	-	-	-	-
Yes	−0.22	0.04	1.22 (1.13–1.32)	0.81 (0.74–0.88)	<0.001
Diabetes					
No	-	-	-	-	-
Yes	−0.13	0.02	1.13 (1.08–1.18)	0.88 (0.84–0.92)	<0.001
Body Mass Index (kg/m^2^)	0.00	0.00	1.00 (1.00–1.01)	1.00 (0.99–1.00)	0.34
Mean Pulmonary Artery Pressure	0.00	0.00	1.00 (1.00–1.00)	1.00 (1.00–1.00)	0.81
Creatinine	−0.19	0.02	1.19 (1.15–1.24)	0.82 (0.79–0.86)	<0.001
Lung Allocation Score at Match	0.00	0.00	1.00 (1.00–1.00)	1.00 (1.00–1.00)	0.001
Mechanical Ventilator Support					
No	-	-	-	-	-
Yes	−0.08	0.04	1.08 (1.00–1.16)	0.92 (0.85–1.00)	0.043
Transplant Type					
Double	-	-	-	-	-
Single	−0.37	0.02	1.40 (1.34–1.46)	0.69 (0.66–0.73)	<0.001
Medical Condition at Transplant					
Not Hospitalized	-	-	-	-	-
Hospitalized, Non-ICU	−0.09	0.04	1.09 (1.02–1.17)	0.91 (0.85–0.98)	0.012
ICU	−0.17	0.05	1.17 (1.08–1.27)	0.84 (0.77–0.92)	<0.001
Steroids					
No	-	-	-	-	-
Yes	−0.08	0.02	1.07 (1.04–1.11)	0.93 (0.89–0.96)	<0.001
Ischemic Time (Hours)	−0.03	0.01	1.02 (1.01–1.03)	0.97 (0.96–0.98)	<0.001
Model Intercept	8.86	0.10			

CI: Confidence Interval; DBD: Donation After Brain Death; DCD: Donation After Circulatory Death; ETR: Event–time Ratio; HR: Hazard Ratio; ICU: Intensive Care Unit; SE: Standard Error.

## Data Availability

The UNOS registry database was used for this study.

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
