# Peer review of "Lung Transplantation Outcomes in Recipients Aged 70 Years or Older and the Impact of Center Volume"

_jcm, 2023, doi:10.3390/jcm12165372_

Round 1
Reviewer 1 Report
Congratulation for your paper.
Impressive numbers with deeply interesting results.
I only have to report two minor corrections:
- line 239: ...tailored post-transplant care to continuously (to) improve posttransplant outcomes.
- line 247: ...the long-term survival of recipients (in) the < 70 years of age subgroup
Author Response
We appreciate the reviewer comments and their time to review our manuscript. We have revised our manuscript accordingly. A point-by-point response to specific critiques raised by the reviewer is provided below.
Reviewer #1:
Comment #1: line 239: ...tailored post-transplant care to continuously (to) improve posttransplant outcomes.
Response #1: Thank you, the correction has been made.
Comment #2: line 247: ...the long-term survival of recipients (in) the < 70 years of age subgroup
Response #2: We have modified the sentences accordingly.
Reviewer 2 Report
I found the manuscript to be quite enjoyable and well-written. However, I did notice a few areas that could use some improvement.
On line 19 of the abstract, it would be helpful to add the word "characteristics" after "Baseline".
In the Statistical Methods/Results section, please include a multivariable Cox regression model to determine the adjusted hazard ratio for mortality in recipients aged 70 years or older. This should take into account potential confounding variables such as donor and recipient characteristics, as well as centre volume.
When it comes to the tables, in the "Variable" column, it would be best to align the writing to the left and use indentation for the subgroups. The Centre text can be quite confusing.
in Table 2, it would be more accurate to change "Days on Waitlist (Day)" to "Waitlist Time (Days)".
On line 251 of the Discussion section, it would be useful to reference the work of Yang et al.
There are a couple of typos on lines 129 and 163, where "creatine" should be corrected to "creatinine".
Author Response
We appreciate the reviewer comments and their time to review our manuscript. We have revised our manuscript accordingly. A point-by-point response to specific critiques raised by the reviewer is provided below.
Reviewer #2:
Comment #1: On line 19 of the abstract, it would be helpful to add the word "characteristics" after "Baseline".
Response #1: Thank you, the correction has been made.
Comment #2: In the Statistical Methods/Results section, please include a multivariable Cox regression model to determine the adjusted hazard ratio for mortality in recipients aged 70 years or older. This should take into account potential confounding variables such as donor and recipient characteristics, as well as centre volume.
Response #2: We agree with the reviewer that it would be beneficial to include a multivariable model to determine the adjusted hazard ratio for mortality in recipients aged 70 years or older. However, when we attempted a multivariable Cox regression, several variables violated the proportional hazards assumption, including the age > 70 variable. Thus, we elected to use accelerated failure time model with Weibull regression. This has now been included as Table 5 in the manuscript. The methods (lines 93-97), results (lines 198-213), and discussion (lines 233-234, 244-247, 278-279) have been revised accordingly to discuss this additional model.
Comment #3: When it comes to the tables, in the "Variable" column, it would be best to align the writing to the left and use indentation for the subgroups. The Centre text can be quite confusing.
Response #3: We have edited the table accordingly for clarity.
Comment #4: In Table 2, it would be more accurate to change "Days on Waitlist (Day)" to "Waitlist Time (Days)".
Response #4: The correction has been made.
Comment #5: On line 251 of the Discussion section, it would be useful to reference the work of Yang et al.
Response #5: Thank you, the reference has been added.
Comment #6: There are a couple of typos on lines 129 and 163, where "creatine" should be corrected to "creatinine".
Response #6: The corrections have been made.
Reviewer 3 Report
Dear Authors,
I read the paper "Lung Transplantation Outcomes in Recipients Aged 70 Years or Older and the Impact of Center Volume" and found exciting insights. The report is well presented in a reasonably smooth style. However, I would suggest some corrections.
1) Line 76-78: I suggest informing readers about the definition of HVC and LVC instead of simply referring to a reference.
2) Line 104: "...the age of _ 70.." remove the extra space
3) Line 117: "Recipients >70 years old were older..." is a truism and should be avoided in the text since it is reported in Tab 2.
4) Line 118: "...to be _ male.." remove extra space.
5) Figure 1: report in the title or the legend what the * means.
6) Figure 2: the KM graph could be visually ameliorated with simple lines in different colours without the surrounding blurs. It does not help the reader who analyses the survival probability.
7) Lines 212-218-223-226-301: "... _ >70..." remove extra spaces.
8) Line 267: "...that _ LVCs..." remove extra space.
Good job.
Author Response
We appreciate the reviewer comments and their time to review our manuscript. We have revised our manuscript accordingly. A point-by-point response to specific critiques raised by the reviewer is provided below.
Comment #1: Line 76-78: I suggest informing readers about the definition of HVC and LVC instead of simply referring to a reference.
Response #1: We agree with the reviewer and lines 76-78 define HVC (35 or more lung transplants in a calendar year) and LVC (less than 35 transplants in a calendar year).
Comment #2:Line 104: "...the age of _ 70.." remove the extra space
Response #2: Thank you, the correction has been made.
Comment #3: Line 117: "Recipients >70 years old were older..." is a truism and should be avoided in the text since it is reported in Tab 2.
Response #3: We agree with the reviewer and have removed this from the text.
Comment #4: Line 118: "...to be _ male.." remove extra space.
Response #4: The correction has been made.
Comment #5: Figure 1: report in the title or the legend what the * means.
Response #5: We agree with the reviewer and have added that * indicates a p value < 0.05 in the figure.
Comment #6: Figure 2: the KM graph could be visually ameliorated with simple lines in different colours without the surrounding blurs. It does not help the reader who analyses the survival probability.
Response #6: We apologize to the reviewer for the lack of clarity of the meaning of the shaded areas. The shaded areas are 95% confidence intervals which reflect the regions where we are 95% certain the true survival curve lies. We feel it is important to include these to indicate the degree of uncertainty of the model. We have added a caption to the figure explaining what the shaded areas are.
Comment #7: Lines 212-218-223-226-301: "... _ >70..." remove extra spaces.
Response #7: The corrections have been made.
Comment #8: Line 267: "...that _ LVCs..." remove extra space.
Response #8: The correction has been made.
Reviewer 4 Report
This is a well-performed and well-written study that adds novel data to the field of lung transplantation. The debate regarding the upper age limit for LTx remains active, in which some centres handle a strict upper age limit, while others transplant patients aged > 65 and > 70 years. This study, demonstrating that overall survival is significantly lower in recipients aged > 70 years, may aid clinicians in making their decision whether to put certain possible candidates on the transplant list or not.
A few minor comments:
- Table 1: would be good to add the definition of pulmonary infection in the table legend (based on positive BAL? then maybe better to say pulmonary infection/colonisation?)
- Table 2: would move sex to 2nd row, under age.
- Section 3.4: would mention the higher incidence of cigarette use and cardiac surgery history too.
- Table 3: it is a bit confusing with the 1y-OS and 3y-OS in this table. I assume this is based on Kaplan-meier curves with log-rank, taking into account duration of FU. Best to specify this in the table legend. Also PGD3: I’m sure the authors mean primary graft dysfunction and not prostaglandin D3.
- Table 3: might be good to give some examples of other COD.
Some typos: line 235 to die of (instead of to) chronic PGD, line 240: to continue to improve instead of continuously, line 241: increased population of LTx recipients, line 283: HVCs being more likely.
Author Response
We appreciate the reviewer comments and their time to review our manuscript. We have revised our manuscript accordingly. A point-by-point response to specific critiques raised by the reviewer is provided below.
Comment #1: Table 1: would be good to add the definition of pulmonary infection in the table legend (based on positive BAL? then maybe better to say pulmonary infection/colonisation?)
Response #1: Pulmonary infection was determined during the donor procurement process where any suspicion of clinical infection is confirmed by culture and reported if positive. We agree with the reviewer and have added this definition in the table legend.
Comment #2: Table 2: would move sex to 2nd row, under age.
Response #2: The correction has been made.
Comment #3: Section 3.4: would mention the higher incidence of cigarette use and cardiac surgery history too.
Response #3: We agree with the reviewer and this has been added in lines 135-137.
Comment #4: Table 3: it is a bit confusing with the 1y-OS and 3y-OS in this table. I assume this is based on Kaplan-Meier curves with log-rank, taking into account duration of FU. Best to specify this in the table legend. Also PGD3: I’m sure the authors mean primary graft dysfunction and not prostaglandin D3.
Response #4: PGD3 has been corrected to primary graft dysfunction, thank you. The 1y-OS and 3y-OS in this table refer to calculations of who was alive/deceased at those time points and does not take into account censoring or follow-up time. The Pearson’s Chi-square test was the statistical test for this comparison. To add clarity, we added to the legend that these are raw calculations of survival, not estimates.
Comment #5: Table 3: might be good to give some examples of other COD.
Response #5: We agree with the reviewer that examples of other COD would be useful, however the UNOS database unfortunately does not provide this information. We apologize to the reviewer that we cannot provide this information. The “other category” is best defined as any cause of death that doesn't fall under one of the aforementioned categories.